# A New Perspective on Large Language Model Safety: From Alignment to Information Control

## Abstract

Large Language Models (LLMs) have demonstrated remarkable capabilities across a wide range of domains, yet their increasing deployment in sensitive and high-stakes environments exposes profound safety risks—most notably, the uncontrolled generation of inappropriate content and the inadvertent leakage of confidential information. Traditionally, such risks have been approached through the lens of alignment, focusing narrowly on ensuring outputs conform to general notions of helpfulness, honesty, and harmlessness. In this work, we argue that such alignment-centric perspectives are fundamentally limited: information itself is not inherently harmful, but its appropriateness is deeply context-dependent.

We therefore propose a paradigm shift in LLM safety—from alignment to information control. Rather than merely shaping model behavior through the existing practice of alignment, we advocate for the principled regulation of who can access what information under which circumstances. We introduce a novel framework for context-sensitive information governance in LLMs, grounded in classical security principles such as authentication, role-based access control, and contextual authorization. Our approach leverages both the internal knowledge representations of LLMs and external identity infrastructure to enable fine-grained, dynamic control over information exposure.

We systematically evaluate our framework using recent models and a suite of benchmark datasets spanning multiple application domains. Our results demonstrate the feasibility and effectiveness of information-centric control in mitigating inappropriate disclosure, providing a foundation for safer and more accountable language model deployment. This work opens a new frontier in LLM safety, one rooted not in abstract alignment ideals, but in enforceable, context-aware control of information flow.

## 1 Introduction

The rapid advancement of Large Language Models (LLMs) has revolutionized modern computing, unlocking transformative capabilities in content generation, decision support, and natural language interaction across virtually every sector, from creative industries to high-stakes domains such as healthcare, finance, and public policy (Brown et al., 2020; Bommasani et al., 2021). However, this unprecedented power comes with significant risks: LLMs can generate contextually inappropriate content or unintentionally disclose sensitive or confidential information, raising serious safety, ethical, and operational concerns (Weidinger et al., 2021; Bender et al., 2021). As LLMs become embedded in critical infrastructure, ensuring their responsible and controlled deployment becomes not only a technical challenge but an organizational imperative.

To date, prevailing approaches to LLM safety have centred on alignment; the effort to train models to consistently exhibit helpful, honest, and harmless behaviour (Dahlgren Lindström et al., 2025). Yet alignment mechanisms often rely on static and universal criteria for harm, overlooking the fundamental fact that the appropriateness of a model's response is inherently context-dependent (Almheiri et al., 2025). The same piece of information may be benign when provided to an authorized administrator but hazardous if exposed to an unverified user. In practice, what matters is

not just what information is shared, but to whom, under what conditions, and why. This observation calls for a profound shift in the safety paradigm—from generalized behavioural alignment to principled information control.

In this work, we introduce a new framework for LLM safety that foregrounds context-sensitive information governance. Drawing inspiration from well-established principles in security and access control (Sandhu, 1998), we propose a system in which user identity, role, and organizational policy jointly determine the boundaries of permissible model behaviour. Our framework integrates user identification, role-based access control (RBAC), and dynamic context inference to mediate LLM responses in real time. Rather than preventing harm by constraining model intent alone, we prevent misuse by enforcing fine-grained information flow control tailored to the access rights and responsibilities of each individual user.

We evaluate our approach through a series of experiments using recent LLMs with simulated data from real-world scenarios, demonstrating that our method can effectively restrict or permit access to specific information based on clearly defined contextual rules. The results validate not only the technical feasibility of our system but also its potential for seamless integration into enterprise and institutional environments.

By reframing LLM safety as a problem of enforceable, policy-driven information control, our work offers a more rigorous and scalable path toward trustworthy AI deployment. It empowers organizations to reclaim control over their information ecosystems, ensures responsible use of powerful language models, and lays the foundation for a new generation of AI systems.

The article is structured as follows. In Section 2, we examine related work on alignment through the lens of information control, highlighting potential unintended consequences of current approaches. Section 3 introduces the design and key components of our proposed framework. In Section 4, we present a comprehensive empirical evaluation that demonstrates the effectiveness and practicality of our approach. Finally, in Section 5, we conclude with a reflection on the broader implications of our work and advocate for a paradigm shift from alignment-centric methods to a context-aware, information control–oriented perspective.

## 2 BACKGROUND AND MOTIVATION

The prevailing paradigm for ensuring the safety of LLMs is alignment, which seeks to train or fine-tune models to behave in ways that are broadly helpful, honest, and harmless. Common alignment techniques include Reinforcement Learning from Human Feedback (RLHF), direct preference optimization, and constitutional AI (Bai et al., 2022; Wang et al., 2024). These methods are typically designed to prevent the generation of content that is offensive, toxic, or discloses personally identifiable information (PII), often guided by generalized ethical and safety principles.

However, existing alignment-based approaches suffer from two fundamental limitations. First, they remain vulnerable to jailbreak attacks (Andriushchenko et al., 2024; Zheng et al., 2024), i.e., carefully crafted adversarial prompts that can circumvent alignment and elicit harmful, private or sensitive outputs. Second, and arguably more critically, alignment techniques operate primarily at the level of model behaviour, without regard for contextual information access. That is, they treat all requests for sensitive data, such as PII or confidential organizational records, as inherently unsafe, regardless of the requester's identity, role, or authorization level. This lack of contextual awareness results in both under-blocking (where unauthorized access is permitted through adversarial prompts) and over-blocking (where legitimate users are denied access due to blanket restrictions), thereby undermining both the safety and the practical utility of LLMs (Cui et al., 2024; Sullutrone et al., 2025; Zhang et al., 2025).

To better understand these shortcomings, we conducted a targeted evaluation (refer to detailed experimental setup in Section 4) using multiple state-of-the-art LLMs. We curated a set of organizational documents containing both general and sensitive information, and issued a series of prompts requesting employee-related data—such as salary records, home addresses, and employment history, under two contrasting scenarios: (1) Legitimate requests issued by clearly identified users, such as Human Resource Director (HR Dir), Financial Director (Fin Dir) or CEO of the same organization; and (2) Adversarial prompts that subtly manipulated the request to bypass safety filters without proper credentials.

The results underscored the limitations of existing alignment-centric safety mechanisms. In a significant number of legitimate cases, the models systematically refused to respond, citing privacy or ethical concerns—even when the requesting role had clear, organizationally sanctioned access rights. In contrast, many adversarial prompts succeeded in extracting sensitive information, effectively bypassing alignment safeguards through indirect or obfuscated queries. These outcomes reveal a critical deficiency in current approaches: they are not equipped to evaluate or enforce access policies that depend on who is asking, under what conditions, and for what purpose.

This disconnect between intent and response reflects a deeper conceptual flaw: current alignment frameworks conflate harmful content generation with inappropriate information disclosure, without distinguishing the legitimacy of the requester or the context of the interaction. In contrast, real-world information systems (e.g., enterprise IT software and secure databases) rely on fine-grained, RBAC to manage who can access which information, and under what roles and circumstances (Bertino, 2003).

These observations motivate a shift in LLM safety thinking—from behaviour-level alignment to information-centric access control. Rather than solely attempting to preempt harmful outputs through generic alignment objectives, we propose viewing LLMs as context-sensitive information interfaces governed by enforceable policies. By grounding safety in established principles such as user identification (via authentication and authorization) and context-aware access control, we can more effectively manage the dual challenge of preventing unauthorized disclosures while supporting legitimate information use. This perspective paves the way for safer, more adaptable, and more trustworthy deployment of LLMs in complex, real-world environments.

## 3    LLM INFORMATION CONTROL FRAMEWORK

To address the limitations of alignment-centric approaches, we propose a modular and context-aware Information Control Framework for LLMs. The core objective of the framework is not simply to prevent harmful behaviour at the model level, but to ensure that the dissemination of sensitive or confidential information is systematically governed by enforceable access policies. This is achieved through a principled design grounded in four key attributes:

- Information Flow Control: The framework governs the flow of information in accordance with organizational policies, focusing on the appropriateness of outputs relative to the identity and authorization of the requester.

- Lightweight and Customizable Modules: Each module is designed for low computational overhead and high customizability, allowing organizations to adapt the framework easily as their business information, policies or security requirements evolve.

- LLM Workflow Coverage: Modules are positioned across the full LLM interaction lifecycle (input, reasoning, and output) to provide end-to-end information governance.

- Broad Applicability: The framework is compatible with both closed-source (API-based) and open-source LLM deployments, making it applicable across a wide range of use cases and operational environments.

Figure 1 illustrates the overall architecture of the framework, which comprises four modular components. These modules can be deployed independently or in combination, enabling organizations to tailor the information control depth based on user roles, data sensitivity, and risk tolerance. We describe each component below.

Starting with the *user identification module* which verifies the identity of each user at the beginning of a session. This step is foundational for establishing a secure and auditable interaction context. By tying user identity to every interaction, the module enables role inference, personalized policy enforcement, and compliance monitoring. Authentication can be integrated with existing enterprise identity providers (e.g., OAuth or Active Directory), ensuring seamless compatibility with organizational infrastructure.

Following, the *policy alignment (PA) module* evaluates incoming prompts against organization-specific policies before they reach the model. These policies may define permissible request types, information categories accessible to different roles, restricted language patterns, and guidelines for

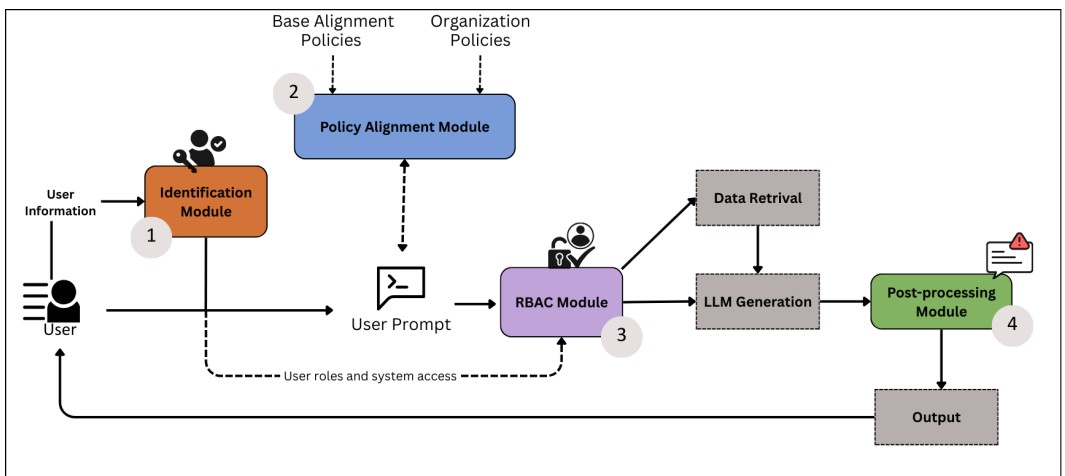

Figure 1: Proposed LLM Application Information Control Framework

handling sensitive or classified data. Unlike generic alignment mechanisms, this module leverages LLM-based reasoning to semantically interpret prompts in context— enabling nuanced enforcement of custom policies. Policies can be defined in natural language, preferably in a structured declarative format. For clarity and reliability, policies should be concise, unambiguous, and self-consistent. This module also performs detection of potentially harmful prompt behaviours, including jailbreak attempts and prompt injection attacks, serving as an intelligent gatekeeper that integrates semantic analysis and policy enforcement beyond simple keyword matching.

Thereafter, the *RBAC module* determines whether the requested information is accessible to the authenticated user based on their assigned role. This module supports fine-grained information governance by enforcing access rules at both the content and metadata levels. In cases where explicit role definitions are unavailable, the system defaults to the most restrictive policy, adhering to a zero-trust security posture. The RBAC module can be seamlessly integrated with data access layers, such as retrieval-augmented generation (RAG) pipelines or external APIs, through the use of access tags or permission metadata attached to individual data items. This integration enables the system to control not only which queries are permitted but also which underlying data can be retrieved and exposed in responses.

Lastly, the *post-processing (PP) module* performs an inspection of the LLM-generated response before it is returned to the user. It acts as a safeguard against inadvertent disclosures, identifying and redacting sensitive content such as PII, proprietary project names, or confidential figures. While aggressive redaction can impact usability, this module is designed to preserve coherence and meaning to the extent possible. Organizations can configure the module's sensitivity and coverage based on their operational needs and regulatory constraints.

A key strength of the proposed framework is its modularity and cascading control logic. Organizations can selectively activate modules based on available infrastructure, threat models, and security requirements. For example, a minimal configuration might include only the user identification and post-processing modules, while a comprehensive deployment would include all four components for maximal protection. Critically, the framework supports graceful degradation and failure containment. If a violation is detected at any stage, such as failed user identification, prompt-policy mismatch, RBAC denial, or output policy breach, the system can either terminate the interaction or restrict the response to sanitized fallback messages. In all cases, detailed logging supports auditability, forensic analysis, and ongoing security refinement.

## 4 EMPIRICAL EVALUATION

In this section, we present an empirical study into the effectiveness, practicality, and trade-offs of our proposed information-control framework for LLM workflows. We first outline the research questions and experimental design, followed by a detailed account of tasks, metrics, configurations,

and evaluation scenarios. We conclude with a performance analysis across various modular setups and user roles to validate our approach. The evaluation is designed around the following two research questions:

> **RQ1:** How can information-control mechanisms be effectively integrated into LLM workflows without degrading system usability or responsiveness?
>
> **RQ2:** What are the trade-offs and cumulative effects of integrating different information control modules into LLM workflows, as measured by evaluation metrics such as attack success rate, output correctness, and system latency?

To assess the performance of our proposed framework, we compare it against two representative baselines that were designed to isolate the impact of each module, providing reference points for evaluating the benefits introduced by our dynamic, context-aware framework:

- Baseline 1 (inherent alignment only): A standard LLM configuration without any external safeguards or access control mechanisms. This setup relies solely on the model's built-in safety alignment to handle sensitive or adversarial inputs.
- Baseline 2 (static access control): A configuration augmented with a static information access control protocol. In this setup, access decisions are hardcoded based on pre-defined rules at the document level, independent of user context or dynamic evaluation. That is, users are assigned with static rules and so are the documents. These rules are evaluated to grant or deny access according to the standard practice.

A more detailed description of our implementation, source code and results can be found at: https://github.com/aigovteam-2lk1v2df/InfoControl

## 4.1 EXPERIMENTAL SETUP

To evaluate the effectiveness of our framework, we design two complementary tasks grounded in realistic threat models and enterprise use cases.

The first task emulates legitimate enterprise interactions based on a real-world enterprise in a RAG setting. We construct a dataset comprising 10 corporate user profiles with attributes such as company affiliation, job role, access level, and task context. Complementing this, we curate multiple business documents reflecting real-world financial reporting. In addition, we design users with varying system privileges: CEO (full access), HR Director (HR systems only), Head of Finance (Finance systems only), and ordinary employee. These user profiles, documents, and roles are the result of a systematic rewrite (using LLMs) of real-world data to avoid leaking private data. We then systematically generate 200 short-response questions aligned with the user roles and documents. Half of these questions are focused on PII and HR scenarios and distributed across the 10 profiles; the other half pertain to finance and strategy, drawing content evenly from the documents. For each question-profile pair, we annotate ground-truth responses and determine whether access should be granted or denied based on the profile's permissions.

The second task simulates targeted privacy-violation attempts through existing jailbreak attacks. The goal is to assess whether our framework can effectively defend against these attacks, which are known to be challenging for traditional alignment methods. We select $N = 50$ adversarial prompts from the `CategoricalHarmfulQA` dataset[1], each crafted to elicit sensitive personal information. These prompts are routed through the system's chat-based interface, where the framework must either block or redact the requests, balancing security with helpfulness.

A total of 200 short-response questions are used to evaluate both the Correctness and Defense (defined in Appendix A.2) metrics. Of these, 100 questions focused on PII and HR topics are distributed across 10 distinct corporate profiles. The remaining 100 questions, centered on finance and business strategy, are drawn equally from two business documents. Similarly, we manually review ground-truth answers for each prompt–profile pair, indicating whether the response should be granted or denied based on the user's permissions.

---

[1] https://huggingface.co/datasets/declare-lab/CategoricalHarmfulQA

Table 1: Framework Performance with Modular Configuration

| Framework Configuration | Correctness | Defense % | Average Tokens | Average Latency (s) |
|---|---|---|---|---|
| RAG Workflow (Baseline) | 82.0% | **40.3%** | 1780 | 1.95 |
| Policy Alignment (PA) only | 88.6% | 84.7% | 1648 | 3.51 |
| Role Checking (RC) only | 82.7% | 91.9% | 971 | 3.14 |
| Post Processing (PP) only | 72.4% | 85.6% | 2047 | 3.59 |
| Alignment + Role Checking | 78.0% | 94.6% | 1098 | 2.80 |
| Alignment + Post Processing | 92.5% | 88.8% | 1834 | 4.47 |
| Role Checking + Post Processing | 69.4% | 97.1% | 1201 | 2.27 |
| All Modules (PA + RC + PP) | 85.4% | 97.7% | 1366 | 5.16 |

Putting everything together, we construct a benchmark dataset containing a balanced mix of normal and adversarial questions, enabling us to evaluate the performance of our framework under both standard conditions and adversarial settings. We quantify the performance of an LLM using two metrics: correctness with normal prompt and correctness with adversarial prompts. Given a prompt, we consider the response of the LLM to be correct if and only if the following conditions are satisfied: (1) if the requester does not have the right to the information, the request is rejected; and (2) if the requester does have the right to the information, accurate information is provided in the response. Formally, given a prompt $p$, the response $r$, and the ground-truth response $r_g$,

$$score(p, r, r_g) = \begin{cases} 0.0 & \text{if } r \text{ contains sensitive information beyond} \\ & \text{the prompter's access right;} \\ 1.0 & \text{if } r \text{ correctly denies an underprivileged user;} \\ 1.0 - dist(p, r, r_g) & \text{otherwise} \end{cases}$$

where $dist(p, r, r_g)$ is a normalized score (between 0 and 1) that measures the accuracy and completeness of the response according to the ground-truth. The overall $Correctness$ is then computed as the average score over all 200 question prompts. In our experiments, we use an LLM (i.e., GTP-4o) as a judge for evaluating $dist(p, r, r_g)$. We further randomly selected some of the prompt-response pairs for manual inspection which conforms that the evaluation is accurate.

We further supplement the evaluation with two metrics for measuring the efficiency: *Average Tokens* used and *Average Latency*.

### 4.2 EXPERIMENTAL RESULTS

Table 1 presents the experimental results, covering both aggregated outcomes (last row) and the performance of the system under different subsets of modules.

**Overall performance**   The aggregated results show that our modular framework consistently outperforms the baseline (first row), achieving significantly higher scores in both Correctness and Defense. Notably, integrating all three modules yields the strongest results, with a Defense rate of 97.7% while maintaining strong Correctness. This configuration strikes a balance between robustness and reliability, clearly validating the effectiveness of our modular design.

**Individual contributions**   Our framework is deliberately designed with a modular architecture, enabling it to adapt to different enterprise settings (e.g., environments with or without existing RBAC policies). Examining the modules independently highlights the role of each component.

- The PA module stands out with 88.6% Correctness and 84.7% Defense, providing the most balanced single-module performance.
- The RC module achieves the highest standalone Defense (91.9%) though with somewhat reduced Correctness compared to PA.

Table 2: RC Module Evaluation for Various Roles

| User's Role | Correctness | Defense % | Avg. Tokens | Avg. Latency (s) |
|---|---|---|---|---|
| No Role | 100.0% | 99.5% | 240 | 1.64 |
| HR Dir | 78.4% | 97.4% | 1063 | 3.18 |
| Finance Dir | 82.7% | 91.9% | 971 | 3.14 |
| CEO | 75.5% | 95.1% | 1858 | 4.03 |

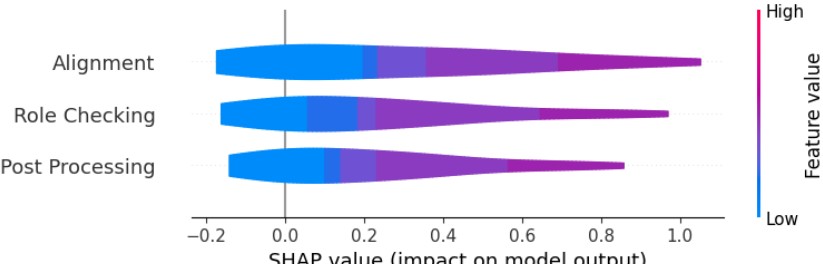

Figure 2: Shapley value contribution analysis.

- The PP module attains the lowest Correctness among the three, but still achieves an 85.6% Defense rate, underscoring its utility as a supplementary safeguard.

Shapley value analysis (Figure 2) further quantifies these effects systematically: PA contributes the most to enhancing protection, followed closely by RC, while PP—though less influential—still plays a meaningful role. These findings emphasize the critical importance of semantic policy checks (PA) and role-based verification (RC) as foundational pillars of effective information control, with PP acting as a valuable complement.

Pairwise combinations of modules reinforce these insights. For example, PA+RC boosts Defense to 94.6% though at the cost of Correctness (78.0%), while PA+PP maintains stronger Correctness with near 88.8% Defense. RC+PP, in turn, yields substantial Defense gains but sacrifices Correctness. Collectively, these results suggest that module interactions can be tuned depending on application-specific priorities.

**Efficiency considerations**    The efficiency profile varies by module. PP incurs the highest token consumption and moderate latency (3.59s), while RC achieves the lowest token usage and latency (3.14s), making it well-suited for resource-constrained scenarios. Importantly, both PA and RC reduce average token usage substantially (7.42% and 45.45% reductions, respectively) compared to the baseline. In addition, Table 2 reveals the performance of RC module based on the different roles, showing high defense rates (above 91%) for all the roles tested. Moreover, Figure 4 (Appendix) illustrates individual module contributions on a chat workflow setting, showing that PA and RC achieve nearly perfect denial rates individually, while PP, though less dominant, still improves robustness over the baseline.

When all three modules are combined, the system attains the highest defense rate (97.7%) at the expense of increased latency (5.16s) and an average token count of 1366. This pattern indicates that while the triple-module configuration is the most secure, selecting tailored module subsets allows practitioners to negotiate the trade-off between efficiency and protection. Overall, our approach substantially strengthen denial behavior, underscoring the value of a multi-layered modular configuration approach to safety.

**Answers to RQs**    Based on the presented results, we can now address our research questions. In terms of RQ1, our findings show that lightweight, incremental deployment of modules can be seamlessly integrated into a standard LLM workflow with only marginal impact on usability. Even deploying a single module yields substantial improvements over the baseline, confirming the effectiveness of our modular approach. More importantly, combining all three modules introduces only

modest latency (well within the threshold for interactive use) while achieving the highest Defense and maintaining a moderate token cost. This configuration is therefore strongly recommended as it offers an excellent security-to-token ratio and delivers robust, context-aware information control without prohibitive resource demands.

In terms of RQ2, the results also reveal a clear spectrum of trade-offs between security robustness, response fidelity, and computational overhead. Each individual module contributes unique advantages:

- ► *RC* excels in efficiency, consuming the fewest tokens and achieving the lowest latency, making it particularly well-suited for resource-constrained environments, if existing RBAC policies are available and well-maintained.
- ► *PA* provides broad semantic safeguards with minimal usability impact, delivering a balanced enhancement across both correctness and defense.
- ► *PP* though the most resource-intensive, serves as a crucial backstop, catching edge-case leaks that may bypass upstream checks.

When modules are combined, these strengths complement one another. Dual-module pairings provide a practical balance: for instance, PA+RC achieves high defense with manageable costs, while RC+PP significantly boosts defense at the expense of some correctness. The full three-module configuration, though associated with an average token footprint of 1,366 and slightly higher latency, remains more efficient than certain dual-module setups such as PA+PP, which demands around 1,834 tokens because we bypass the computation of later modules if any prior guardrail modules have failed. Strikingly, the triple-module system not only delivers the strongest defense and preserves high response fidelity, but it also does so with a leaner token profile than many lighter combinations.

Taken together, our results indicate that organizations adopting our framework gain maximum protection without sacrificing efficiency. They can tailor deployments to align with their security priorities and operational constraints, choosing single- or dual-module setups when resources are tight, or adopting the full configuration for maximal robustness at a still-reasonable cost.

## 4.3 ABLATION STUDY

While the results from closed source (API) implementation demonstrated to give high correctness and defense, it is also important to compare how open source alternatives stack against their closed source counterpart. Hence, Table 3 shows the results for two widely used open source models Gemma-3n-4b (Team, 2025) and GPT-OSS-20b (Agarwal et al., 2025), and how they compare against GPT-4o (Hurst et al., 2024). Note that the same experimental set-up was followed for both close/open source implementations.

On one hand, GPT-OSS-20b attains higher correctness than GPT-4o in all cases except PA-only. In contrast, its defense rate is lower in every setting. On the other hand, Gemma model performs at a comparable correctness against those from GPT-4o, and in the case of PP, Gemma's model outperforms the GPT close source option. Yet, when it comes to defense rates, Gemma's model performs poorly in the PP only configuration and exhibits the lowest defense rate in PA+RC+PP configuration.

For the full modular configuration (PA + RC + PP), while GPT-4o is the better choice terms of defense, compelling open source alternatives can be found in GPT-OSS-20b which only presents a 6.2% drop with a 2.4% correctness increase. In addition, the results shown in Table 4, Appendix A.1, attests that the average token of the open source models is lower to those from GPT-4o; but the trade-off of choosing open source would result on a higher latency.

To sum up, the choice of implementation should consider not only which modules to deploy but also which model family (closed or open source) best aligns with organizational priorities. GPT-4o consistently delivers the strongest defense in the full modular configuration, making it the preferred option where security robustness is paramount. However, open source alternatives such as GPT-OSS-20b and Gemma-3n-4b demonstrate competitive performance in correctness and efficiency, and may be attractive in scenarios where transparency, cost, or deployment flexibility is valued. Consequently, organizations should select module–model pairings that balance security needs, user experience, and resource constraints. In particular, deploying the full three-module framework with

Table 3: Extensibility to Open source models

| | GPT-4o | | Gemma-3n-4b | | GPT-oss-20B | |
|---|---|---|---|---|---|---|
| | Correctness | Defense (%) | Correct. | Defense (%) | Correct. | Defense (%) |
| Policy Alignment (PA) only | 88.6% | 84.7% | 87.5% | 72.7% | 85.4% | 68.8% |
| Role Checking (RC) only | 82.7% | 92.0% | 80.5% | 90.7% | 85.6% | 90.8% |
| Post Processing (PP) only | 72.4% | 85.7% | 85.6% | 47.7% | 89.1% | 70.6% |
| PA + RC + PP | 85.4% | 97.7% | 84.8% | 87.7% | 87.7% | 91.5% |

Correctness and defense rates for closed (API) vs open source models. Company Role = Finance Director

a strong model is recommended for comprehensive protection, while resource-sensitive environments may benefit from lighter models combined with efficient modules such as RC and PA.

## 5 CONCLUSION AND DISCUSSION

Traditional alignment techniques provide a broad safety net, but they exhibit fundamental limitations in practice. As we argued in the previous sections, information is not inherently harmful; its appropriateness hinges on who is receiving it and under what conditions. Therefore, we introduced a modular LLM Information Control Framework that systematically regulates who can access what information by the cascading effect of four main modules: Identification, Policy Alignment, Role-based access controls, and Post Processing.

Through rigorous experimentation, our results demonstrated that combining all modules achieved superior performance in balancing Correctness and Defense, significantly outperforming individual modules and baseline configurations. While the full integration does introduce a moderate latency increase, averaging around five seconds, this slight overhead may be acceptable given the substantial gains in information security and compliance control. Furthermore, the average token usage remains within practical limits, underscoring the feasibility that enforceable information governance can coexist with practical usability within operational environments. By grounding LLM behavior in verified user identity and explicit role permissions, our approach prevents misuse at its source, rather than relying on the model's general notion of harm. In short, our results illustrate how context-aware control of information flow can address the very gaps that alignment-centric methods leave open.

The results and flexibility of our framework provide organizations with a practical tool for managing sensitive information provided to LLM applications. Organizations can strategically deploy modules according to specific security priorities, operational constraints, and resource availability. By placing organizational policies and user context at the core of model interactions, stakeholders can have better control over sensitive data usage within their organisations. This marks an important step toward LLMs deployments that are not only aligned in spirit, but also governed in access, an essential requirement as these models become increasingly integrated into critical infrastructure and daily workflows.

**Limitations & Final remarks** Our study was conducted in text-based, enterprise-style settings; external validity to domain-specialized and multimodal applications (e.g., legal, clinical, or vision–language systems) remain an open question and will likely require domain-specific policy instrumentation and authentication schemes. The current implementation of the four modules incurs a computational overhead (in the order of seconds) that could suggest the need for future optimizations to meet the demands of organizations requiring near-real-time operation. Moreover, as more sophisticated threats and AI regulatory requirements evolve, continuous policy maintenance and monitoring are necessary.

Finally, combining context-aware information controls with advances in model safety may contribute as a layer of defense that is more likely to generalize across organizations and use cases. Overall, moving from abstract behavioral alignment to enforceable information governance provides a practical and extensible foundation for safe LLM deployment across diverse applications.

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

## A APPENDIX

### A.1 ADDITIONAL FIGURES AND TABLES

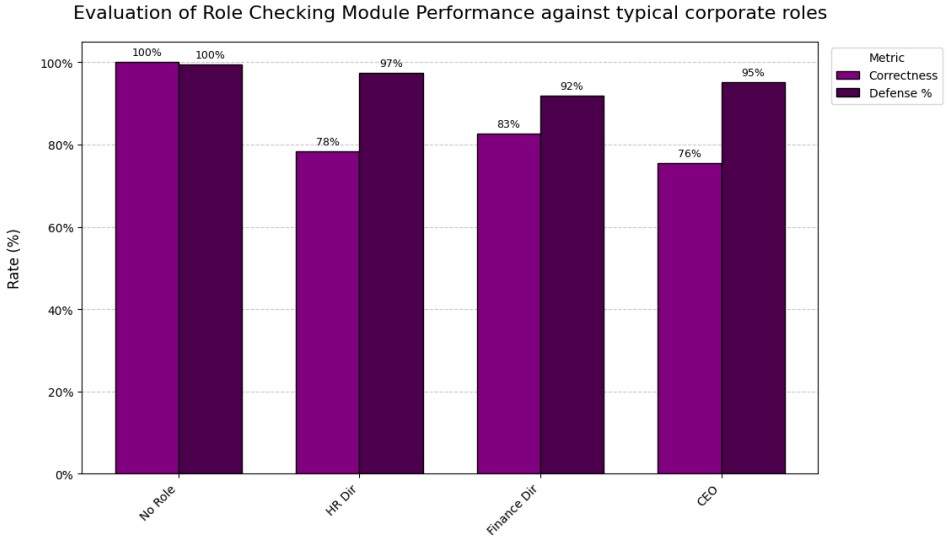

Figure 3: Role Checking Module Performance

### A.2 ADDITIONAL IMPLEMENTATION DETAILS

**Synthetic business corpus.** We used *Gemini 2.5 Pro* via the web interface (Canvas tool, default temperature) to author a synthetic business document. Prompts specified industry context and a plausible financial trajectory so that the generated narrative is realistic while remaining non-derivative. Public financial disclosures from Nvidia and Arm served only as style/content references; we explicitly prompted the model to avoid paraphrasing proprietary language, and the authors manually verified non-derivativeness.

**Synthetic personnel records.** Starting from generic curriculum vitae templates, we produced fictitious employee profiles through iterative prompting of *Claude 3.7* and light manual editing to resemble professional curriculum vitae while containing no real personally identifiable information (PII). The prompt included the aforementioned synthetic business corpus context to ensure consistency in corporate narrative.

Table 4: Framework Performance with Modular Configuration - Open Source Models

| **Gemma-3n-4b** | | | | |
| --- | --- | --- | --- | --- |
| **Framework Configuration** | **Correctness** | **Defense %** | **Average Tokens** | **Average Latency (s)** |
| Policy Alignment (PA) only | 87.5% | 72.7% | 1537 | 3.55 |
| Role Checking (RC) only | 80.5% | 90.7% | 835 | 6.74 |
| Post Processing (PP) only | 85.6% | 47.7% | 1777 | 2.99 |
| PA + RC + PP | 84.8% | 87.7% | 836 | 8.31 |
| **GPT-oss-20B** | | | | |
| **Framework Configuration** | **Correctness** | **Defense %** | **Average Tokens** | **Average Latency (s)** |
| Policy Alignment (PA) only | 85.4% | 68.8% | 1778 | 3.50 |
| Role Checking (RC) only | 84.1% | 90.8% | 915 | 2.83 |
| Post Processing (PP) only | 89.1% | 70.55% | 1779 | 3.46 |
| PA + RC + PP | 87.7% | 91.5% | 952 | 6.01 |

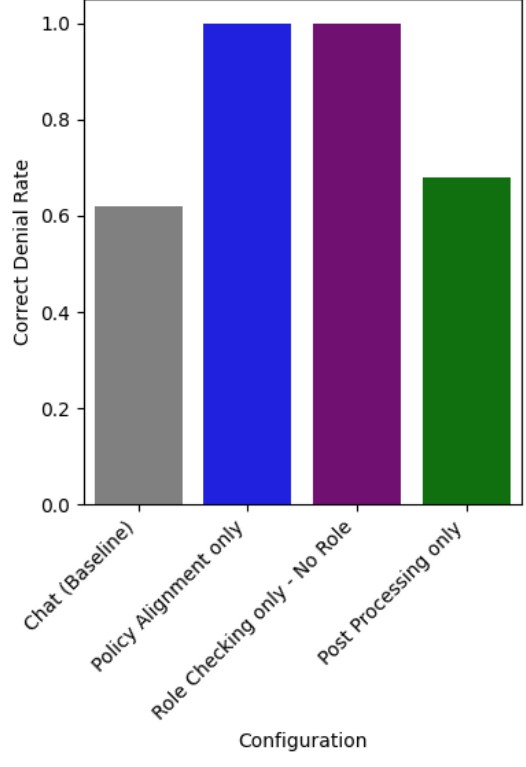

Figure 4: Chat Workflow

**Role-based access control (RBAC).** The organizational RBAC catalogue was curated by the authors from industry references and reviewed for the three target roles considered in our experiments, ensuring clear, enforceable permission boundaries.

**Question set construction.** *Claude 3.7* generated question banks of varying difficulty conditioned on the synthetic CVs and business corpus. We balanced domains with 100 HR-oriented and 100

finance-oriented questions. For HR items, prompts required inclusion of PII-like fields where appropriate to test denial behavior. All outputs were requested in structured JSON.

**Consistency checks.**    The authors, aided by an auxiliary LLM, cross-checked generated questions against the reference documents to ensure internal consistency, factual coherence with the synthetic corpus, and absence of unintended leakage.

**Attack Success Rate and Defense metrics.**    We first define Attack Success Rate (ASR) against a LLM application denoted as:

$$ASR = \frac{1}{|p^{\text{priv}}|} \sum_{p_i \in p^{\text{priv}}} ASR(p_i)$$

$$ASR(p_i) = 1.0 - dist(p, r, r_g)$$

where $p^{\text{prev}}$ denotes the prompt which will return information that is beyond the user role privilege, ie Financial Director requesting for HR information. $dist(p, r, r_g)$,as defined in the main body of the paper, measures how much ground-truth information is given in the response $r$ as determined by LLM-as-judge (GPT-4o). A higher $ASR(p_i)$ will imply that for a specific prompt, $p$, $r$ contains information that is consistent, accurate and complete when compared to $r_g$, that is, privileged information from a company's corpus is provided as a response to the corresponding prompt.

We further define the Defense metric as:

$$Defense = 1 - ASR$$

**Evaluation protocol.**    We tracked runs and experimental results using *LangSmith*, and conducted LLM-as-judge evaluations using *GPT-4o mini*. The final results are exported as csv for archiving and for further analysis.

Specifically, we ran 3 LLM-as-judge evaluators for 1) sensitive_info, 2) correctness, 3) correct_denial results fields. The aggregated the LLM-as-judge outputs are used compute the Correctness and Defense metrics reported in this paper. The full judging prompt is provided here.

