# OpenReview forum: "A New Perspective on Large Language Model Safety: From Alignment to Information Control"
_ICLR.cc/2026/Conference — Submitted to ICLR 2026_

### Official Review · Reviewer_axdv · 2025-10-31

**Soundness:** 2
**Presentation:** 2
**Contribution:** 2
**Rating:** 4
**Confidence:** 2

**Summary:**

This paper proposed an information control framework for large language models, shifting from alignment-based safety to context-aware information flow management. To enhance security, the framework integrates user identification, policy alignment, role-based access control, and post-processing modules. A benchmark dataset was constructed based on real enterprise scenarios, and experiments demonstrated that the proposed method significantly improves defense against unauthorized information leakage while maintaining high correctness and acceptable latency.

**Strengths:**

Unlike existing approaches that primarily focus on content filtering or adversarial training, this research reframes LLM security at the architectural level by proposing "information control" as a new paradigm that offers superior engineering feasibility and organizational adaptability. The technical solution demonstrates rigorous design with clear module segmentation and logical consistency. The inclusion of Shapley value analysis and module combination testing further strengthens the credibility of the conclusions. As LLMs become increasingly integrated into core enterprise workflows, the need for fine-grained, controllable access to sensitive information has become increasingly urgent. This solution that offers significant reference value for both subsequent research and real-world deployment.

**Weaknesses:**

1. Compared to classical rule-based access control systems, the semantic approach of the PA module may introduce new uncertainties and attack surfaces.

2. While the PP module helps prevent information leakage, it may lead to erroneously removing non-sensitive but semantically similar content,  compromising output usability. Although the paper mentions "configurable sensitivity," it does not quantify the usability-security trade-off under different configurations.

**Questions:**

1. The PA module is described as using "LLM-based reasoning to semantically interpret prompts." Is this module itself also an LLM? How does it ensure robustness against the same jailbreaking attacks it aims to prevent? Is its decision-making process auditable?

2. When the PP module incorrectly redacts critical business terminology, how can users obtain actionable feedback?

---

### Official Review · Reviewer_M585 · 2025-10-31

**Soundness:** 2
**Presentation:** 2
**Contribution:** 2
**Rating:** 2
**Confidence:** 4

**Summary:**

The paper proposed a framework that shifts the paradigm of LLM safety mechanisms from reshaping alignment to information/access control. Rather than merely shaping model behavior through the existing practice of alignment, they examined the traditional access control-based technique in LLMs to prevent information leakage.

**Strengths:**

1.	The paper introduces a clear and practical framework that controls who can access what information in an LLM, making it much safer for real-world use in companies handling sensitive data.
2.	The proposed system improves protection against data leaks and harmful outputs while still keeping good accuracy and only a small delay in response time.

**Weaknesses:**

1.	The description of the modules of Figure 1 lacks a complete picture of the methods on how each component works. For example, the user identification module checks if the user is authenticated to have such information. How does this module perform this operation? A detailed methodology is highly required to make it clear.
2.	How would the authors justify the use of GPT-4o as a judge since the same model is used as the target model for evaluation?
3.	Why did the authors only evaluate on those specific three models in the paper, where numerous open-sourced and closed-sourced models are available? What justifies the selection of the specifically 2 OpenAI models and one Google model?
4.	Also, what's the performance of the proposed defense on reasoning-focused models and mixture-of-expert models?
5.	The paper should also test the defense performance against SOTA attacks to understand the proposed defense method’s utility and compare the performance with the other defense techniques under the same attacks.
6.	What are potential failure cases of the proposed method?
7.	This paper severely lacks a performance comparison with the baseline techniques for LLM defense.

**Questions:**

Please follow the Weaknesses.

---

### Official Review · Reviewer_FuvH · 2025-11-01

**Soundness:** 2
**Presentation:** 2
**Contribution:** 2
**Rating:** 2
**Confidence:** 3

**Summary:**

This paper argues that traditional alignment-based safety frameworks (e.g., RLHF, Constitutional AI) are insufficient because they treat all content generation as equally harmful or safe, ignoring who requests the information and in what context.
The authors propose a paradigm shift from alignment to information control, introducing a modular framework inspired by classical access control principles (authentication, RBAC, and contextual authorization).
The framework integrates four modules (user identification, policy alignment, role checking,  and post processing) to regulate LLM outputs based on identity, role, and context.
Experiments on GPT-4o and open-source models show that the combined modules can reach 97.7% defense rate while maintaining high correctness and acceptable latency, outperforming baseline alignment-only and static-control setups.

**Strengths:**

1. The paper provides an original and well-argued conceptual shift: viewing LLM safety as information governance rather than behavioral alignment. This perspective could influence future safety frameworks.
2. The empirical results include comparisons across different module combinations, ablation studies, latency analysis, and both closed- and open-source models.

**Weaknesses:**

1. Although framed as a new paradigm, many components (policy filtering, role-based rules, post-filtering) resemble structured prompt-engineering pipelines rather than a fundamentally new safety algorithm, which raises concerns about how different this really is from prompt conditioning or retrieval gating.
2. While latency and correctness are measured, the paper does not analyze usability degradation, false denials, or context misclassification, which are crucial for deployability.
3. The comparison with alignment-based safety methods is shallow. The paper positions itself as a paradigm shift but doesn't empirically demonstrate where alignment fails and information control succeeds. More comprehensive comparison should be made (such as comparison with other alignment techniques, and representative jailbreak defense methods).

**Questions:**

1. Can this framework be combined with alignment methods to get better results?

---

### Meta-Review · Area_Chair_vPy7 · 2025-12-03

**Summary:**

This paper proposes what is essentially access control rather than alignment from controlling when a model outputs information that could be potentially harmful depending on the setting.  The authors perform empirical evaluation of their method and claim that it can successfully prevent the release of inappropriate model outputs.  Reviewers raised the following significant concerns: (1) the implemented techniques are just prompt engineering which may already exist, (2) the paper does not test out the extent to which their techniques degrade the usefulness of a model, (3) paper purports to be a paradigm shift but does not show where alignment fails and their methods succeed, (4) paper uses GPT4o both as judge and target model which may be problematic, (5) paper didn’t test the defense against SOTA attacks, (6) lacks comparisons to appropriate baselines, (7) the proposed method may also prevent the model from outputting things it SHOULD output too

**Reviewer Concerns:**

No points were addressed since no rebuttals were posted.

**Reviewer Scores:**

The original scores were 2, 2, 4.  Since the authors did not post rebuttals, I assume the reviewers would not change their scores

---

### Decision · Program_Chairs · 2026-01-26

Reject